# Comparison of the Response to the CXCR4 Antagonist AMD3100 during the Development of Retinal Organoids Derived from ES Cells and Zebrafish Retina

**DOI:** 10.3390/ijms23137088

**Published:** 2022-06-25

**Authors:** Yihui Wu, Jin Qiu, Shuilian Chen, Xi Chen, Jing Zhang, Jiejie Zhuang, Sian Liu, Meng Yang, Pan Zhou, Haoting Chen, Keming Yu, Jian Ge, Jing Zhuang

**Affiliations:** State Key Laboratory of Ophthalmology, Guangdong Provincial Key Laboratory of Ophthalmology and Visual Science, Zhongshan Ophthalmic Center, Sun Yat-sen University, Guangzhou 510060, China; wuyh45@mail2.sysu.edu.cn (Y.W.); qiujin@gzzoc.com (J.Q.); chenshlian@mail2.sysu.edu.cn (S.C.); chenx6542@foxmail.com (X.C.); zhangj.c@foxmail.com (J.Z.); zhuangjj7@mail2.sysu.edu.cn (J.Z.); liusan@mail2.sysu.edu.cn (S.L.); yangm59@mail2.sysu.edu.cn (M.Y.); zhoup23@mail2.sysu.edu.cn (P.Z.); haotingchen9464@163.com (H.C.); yukeming@mail.sysu.edu.cn (K.Y.)

**Keywords:** retinal organoid, retinal development, C-X-C chemokine receptor type 4 (CXCR4), zebrafish

## Abstract

Retinal organoids generated from human embryonic stem cells or iPSCs recreate the key structural and functional features of mammalian retinal tissue in vitro. However, the differences in the development of retinal organoids and normal retina in vivo are not well defined. Thus, in the present study, we analyzed the development of retinal organoids and zebrafish retina after inhibition of CXCR4, a key role in neurogenesis and optic nerve development, with the antagonist AMD3100. Our data indicated that CXCR4 was mainly expressed in ganglion cells in retinal organoids and was rarely expressed in amacrine or photoreceptor cells. AMD3100 treatment reduced the retinal organoid generation ratio, impaired differentiation, and induced morphological changes. Ganglion cells, amacrine cells, and photoreceptors were decreased and abnormal locations were observed in organoids treated with AMD3100. Neuronal axon outgrowth was also damaged in retinal organoids. Similarly, a decrease of ganglion cells, amacrine cells, and photoreceptors and the distribution of neural outgrowth was induced by AMD3100 treatment in zebrafish retina. However, abnormal photoreceptor ensembles induced by AMD3100 treatment in the organoids were not detected in zebrafish retina. Therefore, our study suggests that although retinal organoids might provide a reliable model for reproducing a retinal developmental model, there is a difference between the organoids and the retina in vivo.

## 1. Introduction

Organoids generated from human pluripotent stem cells (ESCs or iPSCs) using three-dimensional (3D) cell cultures resemble human tissues, with similar structures, key features, and developmental stages [1]. This technology has been considered a breakthrough achievement of the past decade [2]. An increasing number of current studies reported various organoids ex vivo, such as cerebral, islet, cardiac, and retinal organoids [3,4,5,6]. Organoids are promising tools for species-specific in vitro modeling of development, disease, and therapy, for toxicological studies, and for autologous transplantation therapy for certain diseases [7,8].

The retina is a part of the central neural system, and its development is a complex process that includes the differentiation and migration of retinal precursor cells [9]. Retinal organoids are generated from embryonic stem cells or pluripotent stem cells that are treated according to an elaborate regime with exogenous factors and that predominantly consist of retinal cell types and similar retinal structures [6,9,10,11,12,13]. Moreover, the developmental rate and pattern of retinal organoids are similar to those of embryonic retinas. Many studies have suggested that retinal organoids have the potential to replace the in vivo model for retinal development [2,8,13,14]. For example, a study by Zhong et al. demonstrated that organoids derived from hiPSCs have spatial and temporal features that replicate the development of the human retina in vivo and are characterized by functional photosensitivity [6]. Moreover, retinal organoids produce disease-associated phenotypes in a genetic knockout model of retinitis pigmentosa and X-linked juvenile retinoschisis [15,16]. Thus, there are enormous possibilities for the use of retinal organoids as a source for cell therapies and as a research model.

Human retinal organoids can mimic retinal tissue; however, the lack of inter-organ communication, such as that which takes place in the choroid and vascular systems, is a clear drawback of organoid-based systems [17]. During the development of retinal organoids, all the nutrition is supplied by the medium, so the specification of cell types and development times depend on the protocol that is used [6]. The retinal organoids induced by a previous protocol showed a developmental process similar to that of the in vivo process; i.e., the cell developmental sequence began with the ganglion cells and photoreceptor cells, then the amacrine, bipolar, horizonal and Muller glial cells appeared, and finally the photoreceptor cells gradually matured and obtained the ability to accept light stimulation. Further, the layer structure in retinal organoids was almost the same as the in vivo structure. Some previous research papers indicated great potential for retinal organoids to mimick a disease model, but the previous studies mainly compared the morphological changes and the key cell populations involved in the disease. Few previous studies compared the genetic function in the various cell populations of the organoid and in vivo models, so whether the organoids provided a reliable mimicking model for exploring the genetic function in vitro was not well understood.

To address this issue, we analyzed the developmental process of retinal organoids using the protocol from a previous study [6] and zebrafish retina, after inhibiting CXCR4 with the antagonist AMD3100 [18]. CXCR4 was selected as the target because the mechanism of the effect of CXCR4 on retinal development is relatively clear. For example, CXCR4 is an important factor in regulating retinal neurogenesis; it is expressed in the ganglion cells in zebrafish; and it might be associated with a brief phase of ganglion cell differentiation [19]. In addition, CXCR4 ishighly expressed in retinal stem cells during proliferation rather than differentiation, and it might regulate retinal stem cell proliferation and apoptosis [20,21]. Furthermore, CXCR4 signaling blockage leads to a reduction of ganglion cells in mouse retina [22] and an abnormal retinal axons pathway in the zebrafish retina [23].

Thus, we investigated the changes in cell numbers, the locations of various cells, and neurofilament outgrowth in human retinal organoids and zebrafish retina after treatment with AMD3100 at various time points. The results showed the differences in the developmental patterns of organoids and the retina in vivo. Accordingly, this study provides new insights into the development of retinal organoids ex vivo.

## 2. Results

### 2.1. 3D Retinal Organoids Generated from H9 hESCs Contain Major Retinal Cell Types Arranged in Proper Layers

Organoid formation was performed according to a protocol described previously [6]. As shown in Figure 1A, H9 hESCs were initially cultured in mTesr for embryonic body (EB) generation for 3 days. Then, EBs were seeded and cultured in neural induction medium. After approximately 15 days of induction, the eye field (EF) appeared and developed to the neural retina (NR) after approximately 30 days of induction. The NR was detached and collected for further culture in suspension for retinal organoid generation. To confirm the stemness and expression of CXCR4 in hESCs, H9 hESCs were stained for OCT4, a specific marker of pluripotent stem cells [6] and CXCR4. As shown in Figure 1B, OCT4 was strongly expressed in the nuclei, and CXCR4 was strongly expressed in the cytoplasm of H9 hESCs. After 10 weeks of induction of spontaneous differentiation, the characteristic center-to-periphery wave of neurogenesis and migration to the corresponding retinal layers was observed in frozen organoid sections (Figure 1C). 

MCM2 is a proliferative marker protein that reflects progenitor cells’ proliferation status and is commonly used in staining the retinal progenitor cells in retinal organoids and in vivo retina [24,25]. The MCM2-positive progenitor cells were mostly located in the outer nuclear layer in the organoids. 

HUD is a marker for ganglion cell phenotype. It was colocalized with ISLET1 in the early developmental stages (8 to 13 weeks) of retinal organoids and in vivo retina [26,27]. Ganglion cells (stained for HUD) were located in the center of the organoids. Few intermediary neurons and amacrine cells (stained for AP-2α) close to the central area were observed. Photoreceptor precursors (stained for OTX2) populated the developing outer nuclear layer. NEFL is a marker for neural axons [28], and the axons appeared straight and in succession in the organoids. The structure of the retinal organoids was quite similar to that of the in vivo retina. The organoid contained retinal cell types arranged in a proper layer For example, MCM2-positive retinal precursor cells were located in the inner neuroblastic layer (INBL) and TUJ-1 positive retinal neuron cells were located in the ganglion cell layer (Appendix A).

### 2.2. CXCR4 Location at Various Stages of Organoid Differentiation

CXCR4 is involved in neurogenesis in the retina and brain [23,29]. We assayed the expression pattern of CXCR4 in organoids by double-staining at various periods from 6 weeks to 10 weeks. As shown in Figure 2A, CXCR4 (green) was predominantly colocalized with HuD-positive (red) ganglion cells from 6 weeks to 10 weeks. CXCR4 was expressed in the axons of ganglion cells (white arrows), which is consistent with the expression in fetal retina [28]. Moreover, interneurons, i.e., amacrine cells, originated later than ganglion cells. AP-2α was expressed in a few cells located near the ganglion cell layer (GCL) in the retinal organoids starting at 8 weeks and in higher number of cells in organoids starting at 10 weeks (Figure 2B). OTX2-positive photoreceptors were also detected in retinal organoids starting at 8 weeks (Figure 2C). Most OTX2-positive cells were located in the the periphery of the inner neuroblastic layer. However, CXCR4 was rarely colocalized with amacrine cells or photoreceptors in the organoids (white arrowheads). Thus, these results suggest that CXCR4 plays a key role in the development of RGCs in the organoids at this time point.

### 2.3. Blockade of CXCR4 with the Antagonist AMD3100 Affects the Formation of Retinal Organoids

A previous study demonstrated that the CXCR4 blockade induces morphological changes during brain development [30]. To assay CXCR4′s functions during organoid differentiation, the antagonist AMD3100 was added to the culture medium on day 3 of induction. As shown in Figure 3A, at 6 weeks after differentiation, most retinal organoids consisted of a thick and transparent NR. During subsequent differentiation, the diameter of the organoids increased, and most organoids in both groups maintained transparent NR at 8 weeks after differentiation. However, at 10 weeks after differentiation, most retinal organoids treated with AMD3100 showed nontransparent NRs (green arrowhead), while the retinal pigmental epithelium (RPE) was not detected in the neural retina in vitro (Appendix A). Moreover, as shown in Figure 3B, the diameters of retinal organoids treated with AMD3100 significantly increased, compared with the diameters of the control retinal organoids (6 weeks: PBS, 143.21 ± 24.56, AMD3100, 154.59 ± 19.96; 8 weeks: PBS, 263.61 ± 14.48, AMD3100, 296.37 ± 22.38; 10 weeks: PBS, 303.44 ± 8.31, AMD3100, 396.26 ± 10.79, ** *p* < 0.01). The counting of the retinal organoids indicated that treatment with AMD3100 significantly decreased their formation (PBS: 1; AMD3100: 0.608 ± 0.013; * *p* < 0.05) (Figure 3C).

### 2.4. Blockade of CXCR4 Impairs Retinal Progenitor Cells’ Differentiation in Retinal Organoids

To define the differentiation of retinal progenitor cells after CXCR4 inhibition, the sections of organoids were double-stained with an anti-MCM2 antibody (precursor cells) and an anti-TUJ1 antibody (differentiated neurons). As shown in Figure 4A, the precursor cells were mainly located in the inner neuroblastic layer (INBL), and the differentiated neurons were located in the ganglion cell layer and the peripheral INBL. Then, we compared the ratios of MCM2/DAPI and TUJ1/DAPI in the PBS and AMD3100 treatment groups at various time points. The positive area of MCM2 and TUJ1 was normalized to DAPI by ImageJ. The data indicated that the difference in the ratio of MCM2 to DAPI for the PBS and AMD3100 treatment groups was not significant (6 weeks: PBS, 0.7238 ± 0.0842, AMD3100, 0.7233 ± 0.1264; 8 weeks: PBS, 0.6340 ± 0.0667, AMD3100, 0.6215 ± 0.0734; 10 weeks: PBS, 0.6106 ± 0.0556, AMD3100, 0.6117 ± 0.0531) (Figure 4B). A higher number of TUJ1-positive cells was observed in organoids after differentiation. However, the ratio of TUJ1 was significantly decreased from 8 weeks to 10 weeks after AMD3100 treatment, compared that the ratio in the PBS group (6 weeks: PBS, 0.2016 ± 0.0994, AMD3100, 0.2054 ± 0.1140; 8 weeks: PBS, 0.4295 ± 0.0335, AMD3100, 0.3338 ± 0.0376; 10 weeks: PBS, 0.6744 ± 0.0442, AMD3100, 0.4781 ± 0.0548, * *p* < 0.05) (Figure 4C). Thus, these results suggested that the CXCR4 blockade did not influence retinal progenitor differentiation from ESCs, but impaired retinal progenitor cells differentiated to retinal neurons.

### 2.5. Blockade of CXCR4 Decreases the Ratio of Ganglion Cells and Results in Ganglion Cell Mislocalization in Retinal Organoids

Previous studies demonstrated that CXCR4 is involved in the decreased differentiation of ganglion cells [31]. Thus, retinal organoid sections from 6 weeks to 10 weeks were stained with an anti-HUD antibody, a marker for ganglion cells in the early stage. As shown in Figure 5A, most ganglion cells in the PBS treatment group were located in the central region of the retinal organoids. The proportion of retinal ganglion cells gradually increased as differentiation progressed. Treatment with AMD3100 induced a significant decrease in the ratio of HUD cells, starting at 8 weeks (6 weeks: PBS, 0.2618 ± 0.0468, AMD3100, 0.2592 ± 0.0438; 8 weeks: PBS, 0.4358 ± 0.0575, AMD3100, 0.3332 ± 0.0282; 10 weeks: PBS, 0.4567 ± 0.0282, AMD3100, 0.3095 ± 0.0444, * *p* < 0.05), which was consistent with the ratio of TUJ1 (Figure 5B). Moreover, AMD3100 treatment induced a significant increase in the ratio of ganglion cells located in the peripheral INBL in the organoids from 8 weeks to 10 weeks (6 weeks: PBS, not detected, AMD3100, not detected; 8 weeks: PBS, 0.0071 ± 0.0064, AMD3100, 0.0253 ± 0.0089, * *p* < 0.05; 10 weeks: PBS, 0.0191 ± 0.0038, AMD3100, 0.0498 ± 0.0150, * *p* < 0.05).

Furthermore, the results of the staining for the mature ganglion cell marker Brn3 were consistent with those obtained using an anti-HUD antibody. As shown in Figure 6B, the ratio of Brn3 significantly decreased after AMD3100 treatment (8 weeks: PBS, 0.0543 ± 0.0287, AMD3100, 0.0200 ± 0.0108; 10 weeks: PBS, 0.0932 ± 0.0422, AMD3100, 0.0266 ± 0.0153, * *p* < 0.05, *n* = 15). BRN3 staining in the AMD3100 treatment group also showed more frequent mislocalization at 10 weeks (8 weeks: PBS, not detected, AMD3100, not detected; 10 weeks: PBS, 0.0002 ± 0.0005, AMD3100, 0.0076 ± 0.0030, * *p* < 0.05, *n* = 15) (Figure 6C). These results indicated that the CXCR4 blockade leads to a decrease in the number of ganglion cells and in their mislocalization in the retinal organoids.

### 2.6. Blockade of CXCR4 Leads to a Decrease in Amacrine Cells and Their Mislocalization in Retinal Organoids

Amacrine cells are interneurons of the retina that are located in the ganglion cell layer (GCL) and the inner nuclear layer (INL). As shown in Figure 7A, amacrine cells stained with an anti-AP-2α antibody were detected in the INBL near the ganglion cell layer, and their number increased during the differentiation of the retinal organoids. AMD3100 treatment induced a significant decrease in the ratio of amacrine cells at 10 weeks (8 weeks: PBS, 0.2272 ± 0.0078, AMD3100, 0.0212 ± 0.0093; 10 weeks: PBS, 0.0384 ± 0.0054, AMD3100, 0.0275 ± 0.0048, ** *p* < 0.01, *n* = 10) (Figure 7B). Some AP-2α-positive cells were detected in the periphery (8 weeks: PBS, not detected. AMD3100, 0.0022 ± 0.0018; 10 weeks: PBS, not detected, AMD3100, 0.0069 ± 0.0048, ** *p* < 0.01, *n* = 10). These results showed that the CXCR4 inhibitor decreased the number of amacrine cells and induced their mislocalization in retinal organoids.

### 2.7. Blockade of CXCR4 Results in Mislocalization of Photoreceptor Precursors and Decreases Their Ratio

OTX2 is an important transcription factor in photoreceptor precursor cells [32]. The sections of the organoids were stained with an anti-OTX2 antibody. As shown in Figure 8A, most OTX2-positive cells were detected in the peripheral INBL in the organoids. AMD3100 treatment did not affect the ratio of OTX2 from 8 weeks to 10 weeks (8 weeks: PBS, 0.1004 ± 0.0430; AMD3100, 0.0996 ± 0.0367; 10 weeks: PBS, 0.1739 ± 0.0652, AMD3100, 0.1721 ± 0.0264, *p* > 0.1, *n* = 15) (Figure 8B). However, the CXCR4 blockade induced abnormal ensembles of photoreceptors in the INBL (white arrowheads), similar to cerebral organoids [33]. The ratio of OTX2-positive cells located in the central and subcentral regions was significantly higher in the AMD3100 treatment group 10 weeks after the treatment (8 weeks: PBS, 0.0246 ± 0.001, AMD3100, 0.0336 ± 0.0056; 10 weeks: PBS, 0.0296 ± 0.0114, AMD3100, 0.0837 ± 0.0461, * *p* < 0.05, *n* = 15).

Moreover, we used an anti-RECOVERIN antibody to stain this relatively late marker of photoreceptor precursor cells in the sections of the organoids. Similar to the results shown in Figure 8, RECOVERIN-positive cells were distributed in the peripheral region of the NBL and partly in the GCL and subcentral region of the INBL (Figure 9A). Moreover, AMD3100 treatment resulted in abnormal ensembles, similar to OTX2 staining (white arrow), and a decrease in the ratio of RECOVERIN-positive cells in the organoids at 10 weeks (Figure 9B, 8 weeks: PBS, 0.0900 ± 0.0350, AMD3100, 0.0740 ± 0.0413, *p* > 0.1; 10 weeks: PBS, 0.1681 ± 0.0647, AMD3100, 0.1228 ± 0.0415, * *p* < 0.05). The ratio of RECOVERIN-positive cells located in the central region also increased in the organoids treated with AMD3100 (Figure 9C, 8 weeks: PBS, 0.0217 ± 0.0098, AMD3100, 0.0354 ± 0.0148, *p* > 0.1; 10 weeks: PBS, 0.0319 ± 0.0216, AMD3100, 0.0552 ± 0.0234, * *p* < 0.05). Thus, these results indicated that a CXCR4 inhibitor induced a decrease in the number of photoreceptor precursor cells and their mislocalization in the retinal organoids.

### 2.8. Blockade of CXCR4 Leads to Abnormal Outgrowth of Neuronal Axons in Retinal Organoids

Neural outgrowth plays an important role in physiological retinal function. As shown in Figure 10A, the outgrowth stained with an anti-NEFL antibody from the ganglion cell layer toward the INBL was detected in organoids treated with PBS (white arrows). AMD3100 treatment induced an opposite pathway of the axons (white arrowheads). Moreover, AMD3100 treatment significantly decreased the length of the outgrowth in the retinal organoids (8 weeks: PBS, 0.205 ± 0.055, AMD3100, 0.109 ± 0.041; 10 weeks: PBS, 0.201 ± 0.094, AMD3100, 0.031 ± 0.036, ** *p* < 0.01) (Figure 10B). A significantly higher number of instances of erroneous nerve growth (white arrowheads) were observed in organoids treated with AMD3100 compared with instances of erroneous nerve growth in the control (8 weeks: PBS, not detected; AMD3100, 0.083 ± 0.044. 10 weeks: PBS, 0.030 ± 0.011, AMD3100, 0.084 ± 0.046, * *p* < 0.05) (Figure 10C). Thus, these results indicate that CXCR4 plays a key role in neuronal axon outgrowth in the retinal organoids.

### 2.9. AMD3100 Treatment of Zebrafish Induced Changes in the Retina Similar to Those Detected in Retinal Organoids

A zebrafish model was used to compare the data obtained in the organoids with the results obtained in vivo in the presence of 3 μg/mL AMD3100 added to culture water. At 72 hpf, the CXCR4 mainly expressed in retinal ganglion cell layer (Appendix A), which was constent with a previous study [23]. The 72 hpf zebrafish retinal sections were stained with an anti-acetylated tubulin antibody, a specific marker of zebrafish ganglion cells. As shown in Figure 11A, abnormal outgrowth of the optic nerve was not detected in the retina treated with PBS (white arrows), and AMD3100 treatment induced abnormal distribution of acetylated tubulin (white arrowheads). The number of ganglion cells also decreased significantly, compared to the number in the control (PBS, 73.8 ± 19.718, AMD3100, 50.8 ± 18.256, * *p* < 0.05). Moreover, staining with an anti-calretinin antibody (a marker of amacrine cells) showed that AMD3100 treatment induced few amacrine cells located in the outer nuclear layer (white arrowheads). The ratio of calretinin/DAPI significantly decreased, compared with the ratio in the control (PBS, 0.3311 ± 0.0850; AMD3100, 0.1341 ± 0.0566. * *p* < 0.05). Furthermore, photoreceptor cells were stained with an anti-Recoverin antibody (Figure 11C). The results of the immunofluorescence assays showed that AMD3100 treatment induced a significant decrease in the number of Recoverin-positive cells (PBS, 0.2680 ± 0.0626, AMD3100, 0.1020 ± 0.0156, *p* < 0.05). These results are similar to the data obtained for the retinal organoids, including the decrease in ganglion cells, amacrine cells, and photoreceptors, the distribution of neural outgrowth, and the misplacement of ganglion cells and amacrine cells in the zebrafish retina after AMD3100 treatment; the results are consistent with those of a previous study. CXCR4 played a key role in retinal cell migration and survival and in the nerve pathway in the retina. CXCR4 deficiency impaired the normal retinal development, and the CXCR4 knockdown resulted in a ganglion cell decrease and abnormal neural outgrowth in the zebrafish retina [22,23]. However, we did not detect an induction of abnormal photoreceptor ensembles by AMD3100 in the zebrafish retina.

## 3. Discussion

In the present study, we generated retinal organoids from embryonic stem cells with a formation that was similar to the developmental pattern in vivo [6] and analyzed the effects of the chemo kine receptor CXCR4 on their development. The data indicated that the CXCR4 blockade by an antagonist induced a significant decrease in the formation of retinal organoids, amacrine cells, and ganglion cells; mislocalization of amacrine cells and photoreceptor; and abnormal axonal outgrowth in retinal organoids. Furthermore, these results were compared with the development in vivo in a zebrafish retina model treated with the CXCR4 inhibitor. As it did with the retinal organoids, the CXCR4 blockade decreased the numbers of ganglion and amacrine cells in the zebrafish retina and decreased the number of photoreceptor cells. Normal neural outgrowth was clearly disrupted. However, the mislocalization of of photoreceptors was not observed in the zebrafish retina. Thus, our findings provide morphological evidence of developmental differences between retinal organoids and zebrafish retina in vivo, when treated with an inhibitor of CXCR4.

CXCR4 played a key role in retinal cell migration and survival and in the nerve pathway in the retina. Previous studies demonstrated that CXCR4 deficiency impaired the normal retinal development and CXCR4 knockdown, resultng in abnormal lamination in zebrafish retina [23]. Most of the results on CXCR4′s function in the development of retinal organoids that were obtained in the present study were consistent with the data obtained for the zebrafish retina. For example, the present study’s data of indicated that CXCR4 was predominantly expressed in RGCs and not in amacrine cells or photoreceptor cells in the retinal organoids (Figure 2). This expression pattern of CXCR4 is consistent with the data of a previous study [34], which reported that CXCR4 was only transiently expressed in RGC axons in the organoids. CXCR4 was also expressed in retinal ganglion cells in the zebrafish retina during axonogenesis [23]. Moreover, a number of studies indicated that CXCR4 is an essential factor for the establishment of neural networks in various neuronal systems: i.e., neuronal migration, cell positioning, and axon wiring. The data of the present study also indicated that the pharmacological blockade of CXCR4 induced abnormal neural outgrowth (Figure 9), which is consistent with the results obtained for the zebrafish retina. Figure 11 shows that abnormal distribution of acetylated tubulin (a ganglion cell marker) was observed in the zebrafish retina. A previous study by Li et al. demonstrated that the knockdown of CXCR4 using a morpholine derivative induced the formation of aberrant pathways by retinal axons [23]. Furthermore, the CXCR4 blockade decreased the number of ganglion and amacrine cells in the organoids and zebrafish retina. Thus, CXCR4 may be involved in the neural differentiation of ganglion cells in both organoids and zebrafish retina, suggesting that retinal organoids may be a reliable mimicking model for the neural differentiation of ganglion cells.

In addition, some results obtained in the retinal organoids differed from the data obtained in the zebrafish retina in vivo. First, during organoid development, the blockade of CXCR4 decreased the formation of retinal organoids (Figure 3C). The diameters of the organoids were significantly larger than the diameters in the control group (Figure 2B). However, our data and data from a previous study indicated that interference with CXCR4 did not affect retinal thickness, except for increasing INBL in the zebrafish retina (Figure 11, [23]). Second, the CXCR4 blockade induced changes in cell locations in the organoids and the zebrafish retina. In the organoids, the blockade of CXCR4 induced significant mislocalization of ganglion, amacrine, and photoreceptor cells at week 10. Some ganglion and amacrine cells migrated to the outer layer (Figure 5, Figure 6, Figure 7 and Figure 8). Specifically, the blockade of CXCR4 induced the formation of clumps of cells of photoreceptors in the INBL in the organoids. However, abnormal clumps of cells were not observed in the ONLs in the zebrafish retina after AMD3100 treatment. Recoverin-positive cells were located in the ONLs in the zebrafish retina (Figure 11). We speculate that these discrepancies could have been induced by the surrounding special structures. For example, a pigment epithelial layer was detected in the retina in vivo. Moreover, development in vivo is influenced by the microenvironment, such as the immune and vascular systems.

Comparison of the previously reported CXCR4 function in brain neural cells indicated that some results obtained in retinal organoids were consistent with the data obtained in vivo but not in vitro. The study of Zhu et al. indicated that CXCR4 promoted the survival of human neural progenitor cells by protecting the cells from apoptosis in vitro [21]. However, many in vivo studies reported that CXCR4 regulates neuronal migration by mediating cellular communication during CNS development and does not influence neural progenitor cells [28,30,35]. Our results showed that the ratio of MCM2-positive progenitor cells was not significantly changed after the CXCR4 blockade in retinal organoids (Figure 4). The staining of photoreceptor progenitor cells with OTX2 was not influenced by the CXCR4 blockade (Figure 8B). Furthermore, CXCR4 blockage in retinal organoids resulted in additional cell apoptosis (Appendix A).Thus, the consistency of CXCR4 function in progenitor cells in organoids and brain tissue suggests that the early developmental stages in retinal organoids and in in vivo are similar.

In addition, our data provided four ideas for future studies. First, previous studies demonstrated that the AMD3100 is strictly confined to CXCR4 [36,37]. In addition, AMD3100 was widely used to block the CXCR4 signaling in a previous study [38]. However, AMD3100 might still have an additional target. Further investigations of the AMD3100 off-target effect are required. Second, the blockage of CXCR4 led to cellular mislocalization in the retinal organoids and the zebrafish; a previous study demonstrated that CXCR4 affected the cell migration and tissue polarity by interacting with CXCL12 [39], but the mechanism by which CXCR4 led to the cellular mislocalization in retinal organoids was still not clear. Third, in our study, we only chose similar developmental stages in zebrafish and retinal organoids for analysis, but additional developmental stages should be compared in retinal organoids and zebrafish in the future. Finally, CXCR4 blockage might alter organoid polarity, but how CXCR4 blockage affected the key factors of polarity, such as OLM, remained not clear.

## 4. Materials and Methods

### 4.1. hESC Culture

The hESC line hESC-H9 [40] (WiCell, Madison, WI, USA) was used in the present study. All cell lines were obtained with a verified normal karyotype and were free from contamination. hESCs were maintained in mTeSR1 medium on Matrigel (growth factor-reduced; BD Biosciences, San Jose, CA, USA)-coated plates (STEMCELL Technologies, Vancouver, BC, Canada), according to WiCell protocols. Cells were passaged every 5 to 7 days at approximately 80% confluence. Clearly visible differentiated cells were marked and mechanically removed before passaging.

### 4.2. Differentiation of hESC into 3D Retinal Organoids

The protocol for the formation of 3D retinal organoids was previously published by Zhong et al. [6]. In brief, hESCs were dissociated into small colonies and cultured in mTeSR1 medium (STEMCELL Technologies, Vancouver, BC, Canada), supplemented with 10 μM blebbistatin (Sigma, St Louis, MO, USA). This time point was defined as day 0 (D0) of differentiation. Then, the neural induction medium (NIM) was prepared. The NIM contained DMEM/F12 (1:1), 1% N2 supplement (Invitrogen, Waltham, MA, USA), 1 minimum essential media-non essential amino acids (NEAAs), and 2 mg ml 1 heparin (Sigma). Cell clumps were cultured in suspension on D1 in medium with mTeSR1/NIM at a 3:1 ratio and on D2 in mTeSR1/NIM at a 1:1 ratio. On D3 of differentiation, cell aggregates were reattached on Matrigel-coated culture dishes containing NIM medium. On D16, the reintal developmental medium (RDM) was prepared; RDM contained DMEM/F12 (3:1), supplemented with 2% B27, 1 NEAA, and 1% antibiotic–antimycotic. Then, the clusters were maintained in the RDM medium, and the medium was changed every 2 to 3 days. On D28 of differentiation, horseshoe-shaped neural retina (NR) structures were identified and isolated with a needle under an inverted microscope. Then, the NR structures were collected and transferred to the RC2 medium for long-term suspension culture. The composition of the medium was reported in a previous study [6].

For the CXCR4 blockade, the CXCR4 antagonist AMD3100 (Sigma, St Louis, MO, USA) was added to the medium at 2 μg/mL on D10. The concentration of AMD3100 was confirmed by CCK-8 and migration assays. hESC-H9 treated with 2 μg/mL AMD3100 showed significantly lower migration, and cell viability was not impaired (Appendix A). The retinal organoids were collected for analysis from 6 weeks to 10 weeks.

### 4.3. Immunofluorescence Analysis

Immunofluorescence assays of hESCs were carried out according to a standard protocol. hESCs were fixed with 4% paraformaldehyde for 15 min. Then, 0.1% Triton X-100 (Sigma, St Louis, MO, USA) was used to permeabilize hESCs for 10 min, and 10% normal goat serum was used to block the cells for 30 min. Then, rabbit anti-OCT4 (1:100, Abcam, Cambridge, UK) and rabbit anti-CXCR4 (1:100, Abcam) antibodies were used for staining. DAPI was used to stain the nuclei.

Immunofluorescence assays of the retinal organoids were carried out according to a standard protocol. The retinal organoids were fixed with 4% paraformaldehyde for 30 min, dehydrated in a sucrose gradient, and embedded in optimal cutting temperature (OCT) compound overnight at −80 °C. The cups were sectioned at a thickness of 10 μm using a Leica microtome. After permeabilization with 0.1% Triton X-100 (Sigma, St Louis, MO, USA) for 10 min and blocking with 10% normal goat serum for 30 min, the sections were stained with rabbit anti-HUD (1:100, Abcam), rabbit anti-OTX2 (1:100, Abcam), rabbit anti-NEFL (1:100, Abcam), rabbit anti-MCM2 (1:100, Abcam), mouse anti-TUJ1 (1:100, Abcam), mouse anti-BRN3 (1:100, Santa Cruz, Dallas, TX, USA), rabbit anti-RECOVERIN (1:400, Millipore, Burlington, MA, USA), mouse anti-AP-2α (1:50, DSHB, Iowa City, IA, USA) antibodies, rabbit anti-Cleaved-Caspase 3 (1:100, CST, Danvers, MA, USA), rabbit anti-Ki67 (1:100, CST), or mouse anti-CRALBP (1:500, Abcam). DAPI was used to stain the nuclei.

The zebrafish embryos were stained according to a standard protocol; in brief, the zebrafish embryos were fixed with 4% paraformaldehyde overnight at 4 °C. The samples were dehydrated with sucrose gradient and embedded in an OCT compound overnight at −80 °C. The embryos were sectioned at a thickness of 10 μm using a Leica microtome. After permeabilization with 0.1% Triton X-100 (Sigma) for 10 min and blocking with 10% normal goat serum for 30 min, the sections were stained with mouse anti-Acetylated tubulin (1:100, Santa Cruz), goat anti-calretinin (1:400, Millipore), rabbit anti-CXCR4 (1:200, Genetex, Irvine, CA, USA), or rabbit anti-Recoverin (1:400, Millipore) antibodies. DAPI was used to stain the nuclei.

### 4.4. Quantitative Analysis of Immunofluorescence Staining

To determine the ratio of differentiated cells in each group, the entire organoid section images were captured at 20× magnification from at least ten organoids in each group and at least three sub-fields in each organoid were selected for analysis. Then, ImageJ software was used to quantify the overlay area of DAPI and differentiated cell markers, including TUJ1, MCM2, HUD, BRN3, AP-2α, OTX2, RECOVERIN, NEFL, Acetylated tubulin, and Calretinin. The ratio of these markers to DAPI in the various groups was used for analysis.

To define the mislocalization of ganglion cells, the organoid was divided into a ganglion cell layer and an inner neuroblastic layer (INBL); the ratios of ganglion cells located in INBL to DAPI in the PBS and AMD3100 treatment groups were compared. For the location of amacrine, the ratio of amacrine located in the lateral edge to DAPI was quantitated. To determine the location of the photoreceptor, the ratio of the photoreceptor not located in the outer layer of the organoid (shown inside the dotted line in Figure 8 and Figure 9) to DAPI was compared in each group.

### 4.5. Migration Assay

The migration assay was performed using Transwell inserts (Corning Inc., NY, USA; pore size 8 μm) in 24-well plates. Prior to the assay, hESCs were pretreated for 24 h in mTesr medium with or without 0.5 μg/mL, 1 μg/mL, or 2 μg/mL AMD3100. Then, 1 × 10^5^ cells in 200 μL of mTesr medium were placed in the upper chamber, and 500 μL of the same medium containing 10% FBS was placed in the lower chamber. Plates were incubated for 48 h at 37 °C in an atmosphere of 5% CO_2_; the culture medium was removed, and the filters were washed twice with phosphate-buffered saline (PBS). The cells on the upper side of the filters were removed with a cotton-tipped swab. The cells on the underside of the filters were fixed in 4% PFA for 10 min and stained with DAPI for 15 min. The cells that migrated from the upper to the lower side of the filters were counted in 10 random fields at a magnification of ×40. The assay was performed in triplicate.

### 4.6. Zebrafish

Adult and embryonic zebrafish were maintained according to standard protocols [41] in accordance with Canadian Council for Animal Care (CCAC) guidelines. Embryos were grown at room temperature (RT) of 28.5 °C in embryo medium (EM) and evaluated according to standardized morphological criteria [42]. Starting from 0 h post-fertilization (hpf), the zebrafish embryos were treated with 3 µg/mL AMD3100, which did not influence the survival rate of the embryos [43], or with PBS. The embryo medium was changed every 24 h. All zebrafish were collected for morphology analysis at 72 hpf.

### 4.7. Statistical Analysis

The data are expressed as the mean ± SD. The differences between the mean values were evaluated by a two-tailed Student’s *t*-test (for 2 groups) or analysis of variance (ANOVA; for >2 groups). All calculations and statistical tests were performed by Microsoft Excel 2003 (Microsoft, Redmond, WA, USA) or SPSS 11.5 (SPSS, Chicago, IL, USA) software. *p* < 0.05 was considered significant for all analyses.

## 5. Conclusions

Our results show the developmental differences of retinal organoids and retina in vivo induced resulting from the blockade of CXCR4. Thus, the present study demonstrated that retinal organoids are a useful mimicking research model and revealed the limitations of the developmental models.

## Figures and Tables

**Figure 1 ijms-23-07088-f001:**
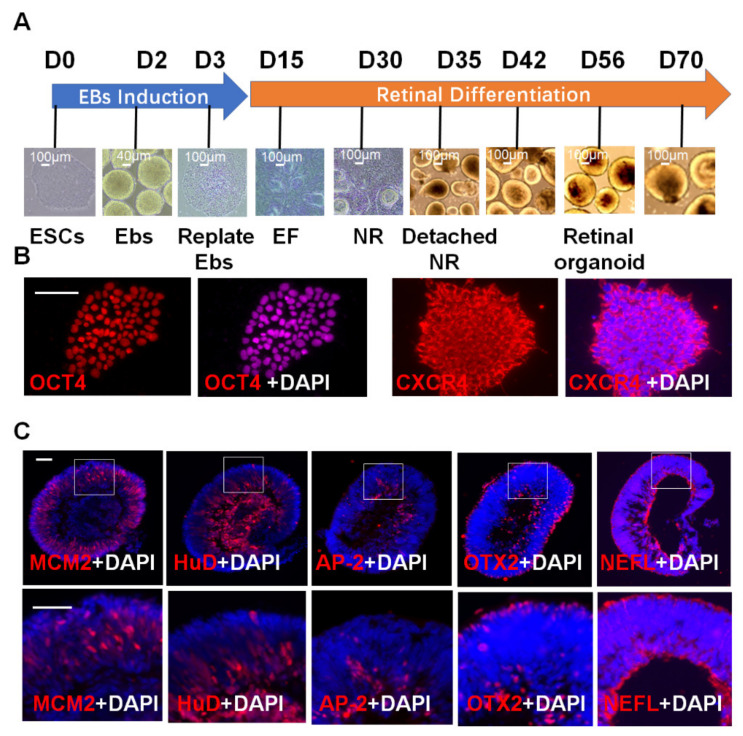
**3D retinal organoids generated from H9 hESCs contain major retinal cell types arranged in proper layers.** (**A**) A scheme of retinal organoids generated by hESCs H9. (**B**) hESCs H9 were stained for OCT4 and CXCR4. (**C**) After 10 weeks of differentiation, retinal organoids contained major cell types of retinal cells, including progenitor cells (MCM2-positive), ganglion cells (HuD-positive), amacrine cells (AP-2α-positive), and photoreceptor cells (OTX2-positive), and showed a normal axon pathway (NEFL staining). Scale bars: 100 μm (**B**,**C**).

**Figure 2 ijms-23-07088-f002:**
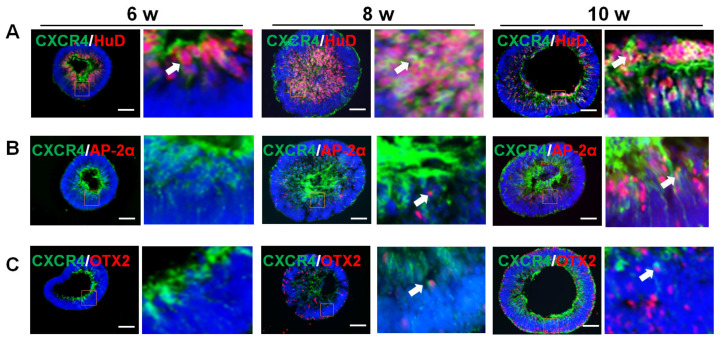
**CXCR4 location during various stages of organoid differentiation.** (**A**) Most ganglion cells stained with HUD (red) were also stained with CXCR4 (green) at various stages of the differentiation of retinal organoids (white arrowhead). (**B**) Few amacrine cells stained for AP-2α (red) expressed CXCR4 (green) during the differentiation of retinal organoids (white arrowhead). (**C**) Rare photoreceptors stained for OTX2 (red) were stained for CXCR4 (green) during the differentiation of retinal organoids (white arrowhead). Scale bars: 100 μm.

**Figure 3 ijms-23-07088-f003:**
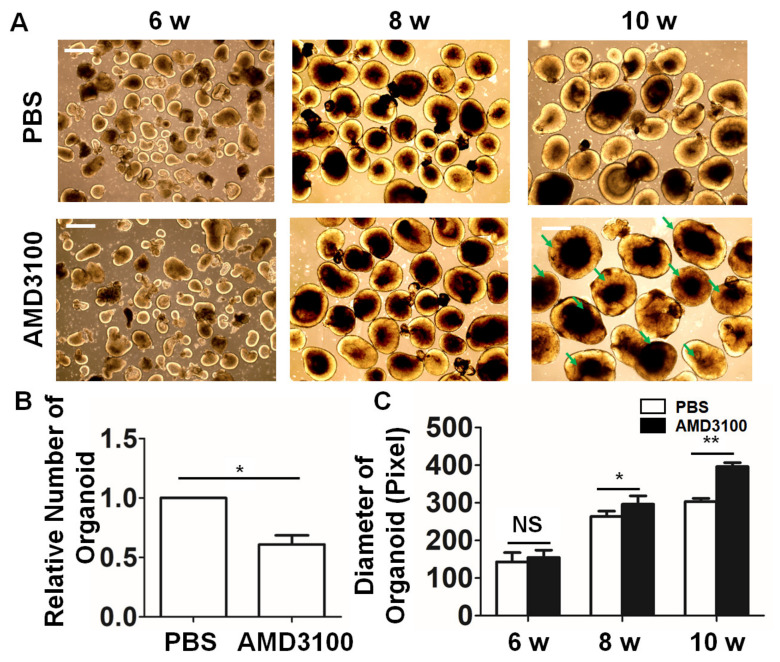
**Blockade of CXCR4 with the antagonist AMD3100 affects the formation of retinal organoids.** (**A**) Images of retinal organoids in the PBS and AMD3100 treatment groups during differentiation. Most retinal organoids treated with AMD3100 showed nontransparent NRs at 10 weeks (green arrowhead). (**B**) The relative numbers of organoids were evaluated in the PBS and AMD3100 treatment groups, and the generation ratio of the organoids significantly decreased (* *p* < 0.05, three independently repeated inductions of retinal organoids were counted). (**C**) The assay of organoid diameters showed that AMD3100 treatment resulted in larger diameters (* *p* < 0.05, ** *p* < 0.01, NS: Not Significant, *p* > 0.05, *n* = 120). Scale bars: 100 μm.

**Figure 4 ijms-23-07088-f004:**
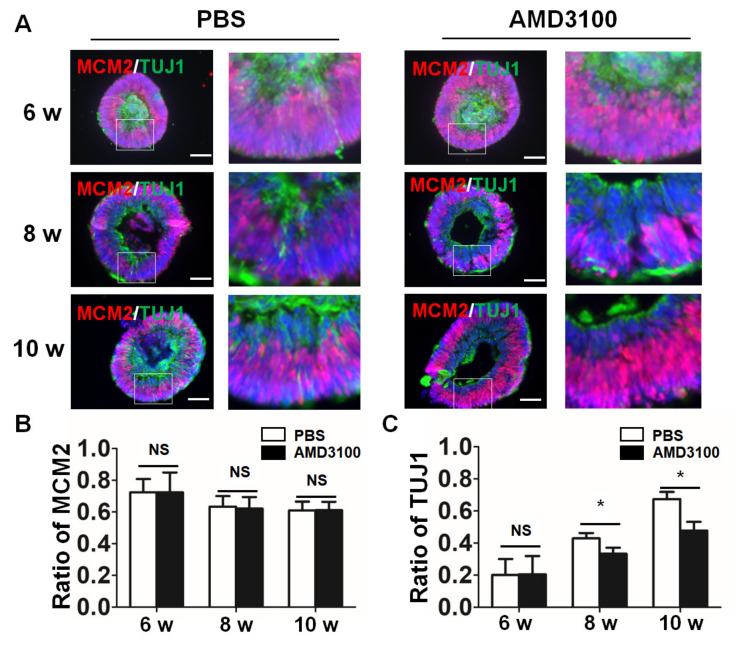
**CXCR4 blockade impairs retinal progenitor cells differentiation in retinal organoids.** (**A**) Double-staining for MCM2 (progenitor cell marker) and TUJ1 (neuron cell marker) in the PBS and AMD3100 groups at various developmental stages of retinal organoids. (**B**) The relative ratio of MCM2-positive cells to all cells (nuclei were stained with DAPI) was not significantly different between the two groups (*p* > 0.1, *n* = 15). (**C**) The relative ratio of TUJ-positive cells to all cells (nuclei were stained with DAPI) was significantly decreased in the AMD3100 group (* *p* < 0.05, NS: Not Significant, *p* > 0.05, *n* = 15). Scale bars: 100 μm.

**Figure 5 ijms-23-07088-f005:**
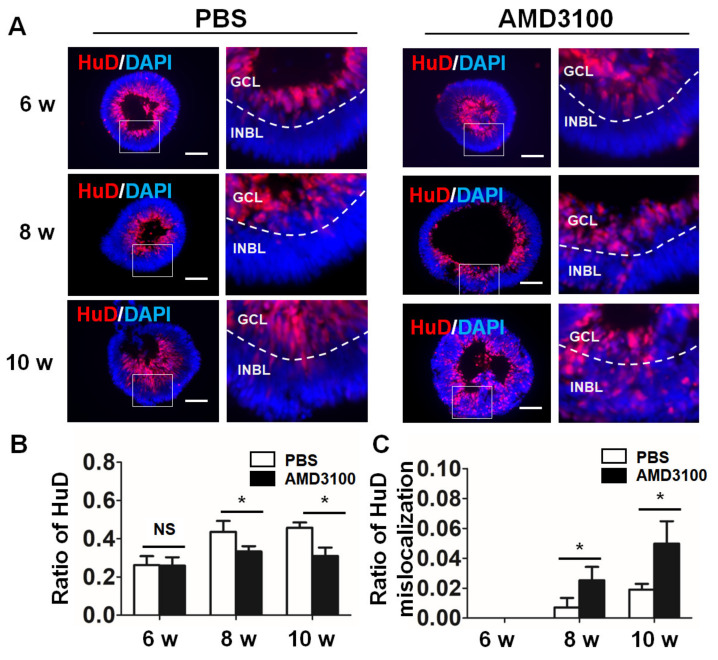
CXCR4 blockade decreases the ratio of early ganglion cells and results in their mislocalization in retinal organoids. (**A**) Ganglion cells were stained by HUD from 6 weeks to 10 weeks to detect abnormal location of HUD-staining in the AMD3100 treatment group (white arrowhead). (**B**) The relative ratio of HUD-positive cells to all cells (nuclei were stained with DAPI) significantly decreased in the AMD3100 group (* *p* < 0.05, NS: Not Significant, *p* > 0.1, *n* = 15). (**C**) The relative ratio of HUD-positive cells to all cells (nuclei were stained with DAPI) located in the INBL significantly increased in the AMD3100 group (* *p* < 0.05, *n* = 15). Scale bars: 100 μm.

**Figure 6 ijms-23-07088-f006:**
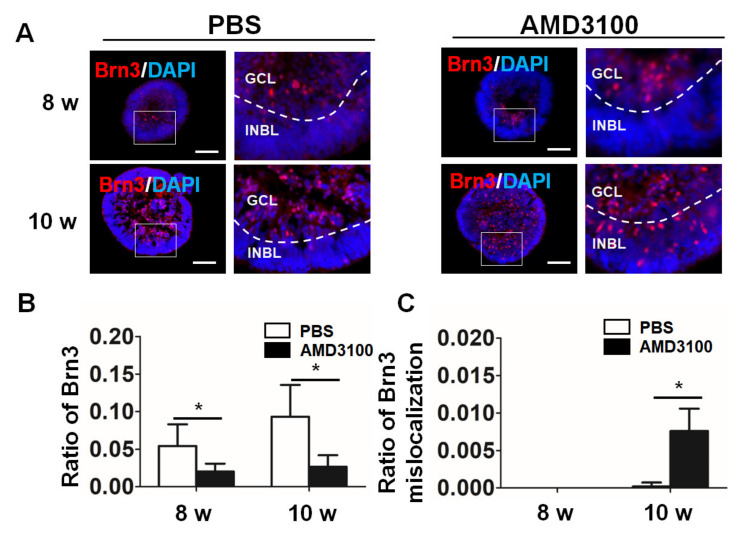
CXCR4 blockade decreases the ratio of mature ganglion cells and results in their mislocalization in retinal organoids. (**A**) Ganglion cells were stained for BRN3 (marker for mature ganglion cells) from 6 weeks to 10 weeks, and the staining of BRN3 was mislocalized in INBL in the AMD3100 treatment group. (**B**) The relative ratio of BRN3-positive cells to all cells (nuclei were stained with DAPI) significantly decreased in the AMD3100 group (* *p* < 0.05, *n* = 15) at 10 weeks. (**C**) The relative ratio of BRN3-positive cells to all cells (nuclei were stained with DAPI) located in the INBL significantly increased in the AMD3100 group (* *p* < 0.05, *n* = 15) at 10 w. Scale bars: 100 μm.

**Figure 7 ijms-23-07088-f007:**
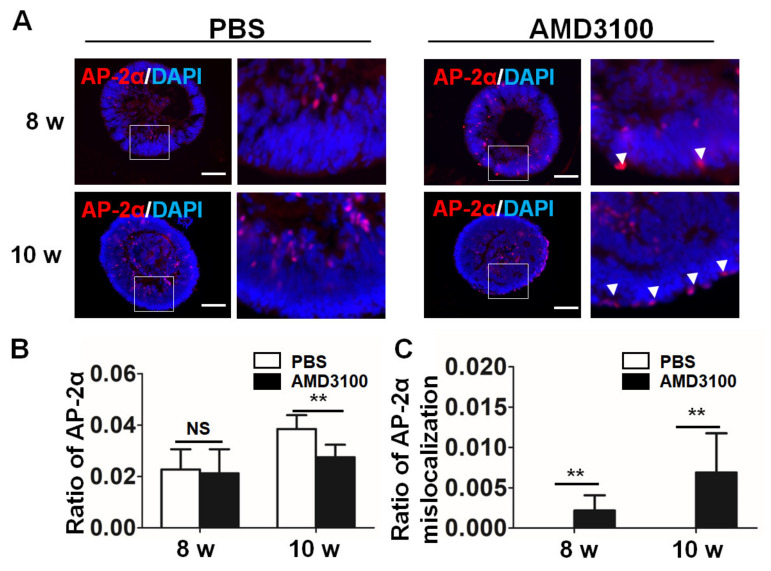
CXCR4 blockade leads to a decrease in the number of amacrine cells and their mislocalization in retinal organoids. (**A**) Amacrine cells were stained for AP-2α from 8 weeks to 10 weeks of differentiation, and AP-2α staining was mislocalized in the AMD3100 treatment group (white arrowhead). (**B**) The relative ratio of AP-2α-positive cells to all cells (nuclei were stained with DAPI) significantly decreased in the AMD3100 group (** *p* < 0.01, NS: Not Significant, *p* > 0.1, *n* = 15). (**C**) The relative ratio of AP-2α-positive cells to all cells (nuclei were stained with DAPI) located in the INBL significantly increased in the AMD3100 group (** *p* < 0.01, *n* = 15). Scale bars: 100 μm.

**Figure 8 ijms-23-07088-f008:**
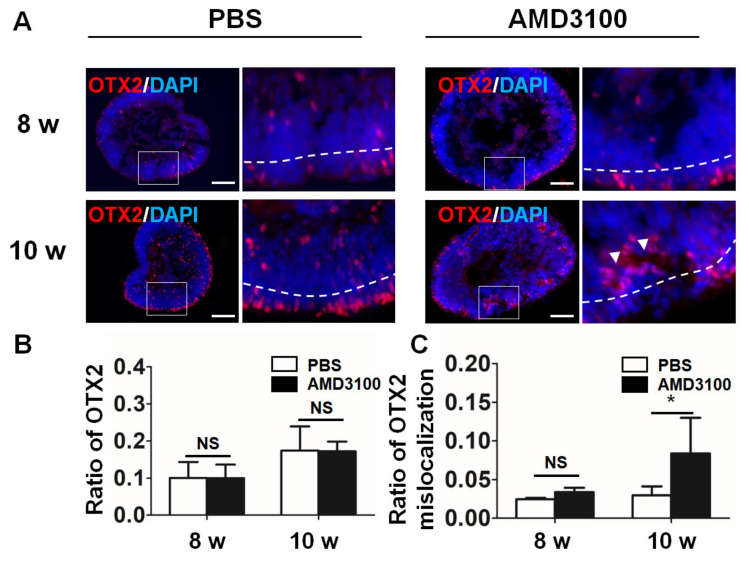
**CXCR4 blockade results in mislocalization of photoreceptor precursors.** (**A**) Photoreceptor precursors were stained for OTX2 from 8 weeks to 10 weeks, and the OTX2 staining was mislocalized in the AMD3100 treatment group (white arrowhead). (**B**) The relative ratio of OTX2-positive cells to all cells (nuclei were stained with DAPI) was not different between the PBS and AMD3100 group (NS: Not Significant, *p* > 0.1, *n* = 15). (**C**) The relative ratio of OTX2-positive cells to all cells (nuclei were stained with DAPI) in the INBL cluster (inside the dotted line) significantly increased in the AMD3100 group (* *p* < 0.05, NS: Not Significant, *p* > 0.1, *n*=15). Scale bars: 100 μm.

**Figure 9 ijms-23-07088-f009:**
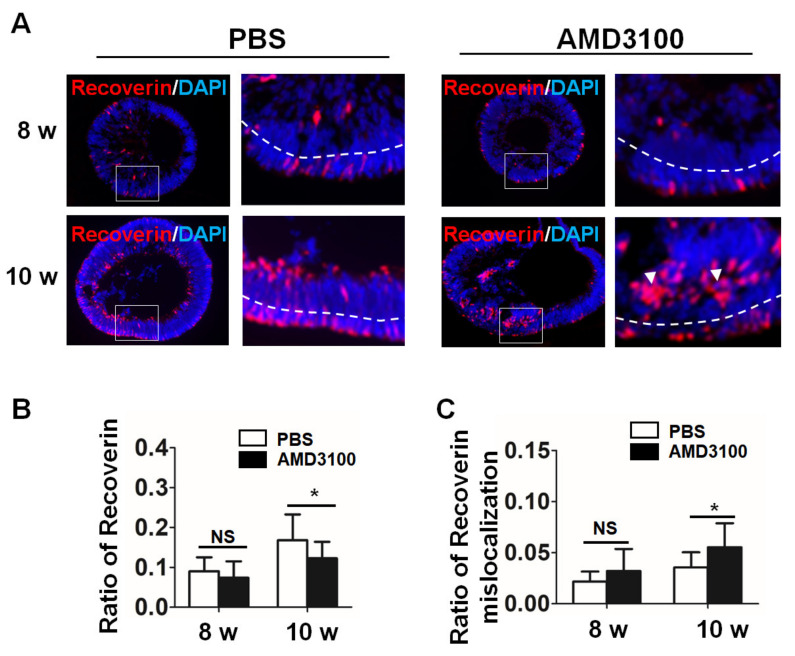
**CXCR4 blockade leads to a decrease in the number of photoreceptor precursors and their mislocalization.** (**A**) Photoreceptor precursor cells were stained for RECOVERIN from 8 weeks to 10 weeks of differentiation, and RECOVERIN staining showed mislocalization in the AMD3100 treatment group (white arrow). (**B**) The relative ratio of RECOVERIN-positive cells to all cells (nuclei were stained with DAPI) significantly decreased in the AMD3100 group (* *p* < 0.05, NS: Not Significant, *p* > 0.1, *n* = 15). (**C**) The relative ratio of RECOVERIN-positive cells to all cells (nuclei were stained with DAPI) in the INBL cluster (inside the dotted line) significantly increased in the AMD3100 group (* *p* < 0.05, NS: Not Significant, *p* > 0.1, *n* = 15). Scale bars: 100 μm.

**Figure 10 ijms-23-07088-f010:**
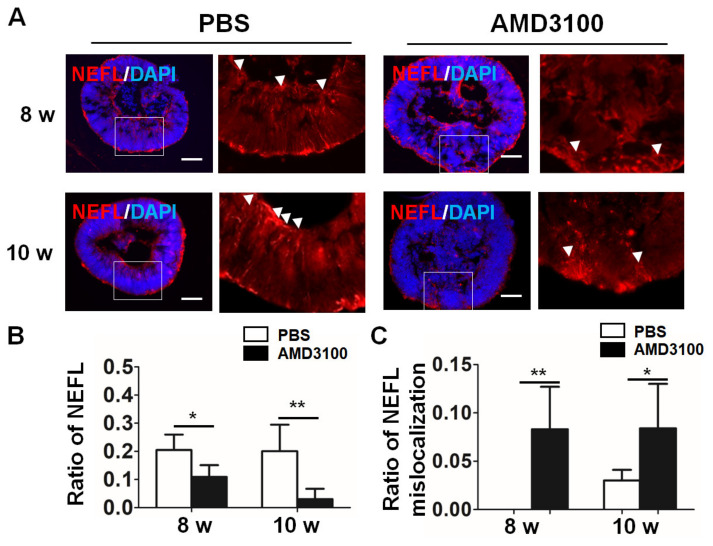
**CXCR4 blockade leads to abnormal neuronal axon outgrowth in retinal organoids.** (**A**) Axons were stained with NEFL in the PBS and AMD3100 treatment groups from 8 weeks to 10 weeks of differentiation, and axons showed an opposite pattern (AMD3100) compared with that in the PBS treatment group (white arrowheads). (**B**) The relative ratio of NEFL-positive cells to all cells (nuclei were stained with DAPI) significantly decreased in the AMD3100 group ((* *p* < 0.05, ** *p* < 0.01, *n* = 10). (**C**) The relative ratio of erroneous NEFL pathfinding to all cells (nuclei were stained with DAPI) significantly increased in the AMD3100 group (* *p* < 0.05, ** *p* < 0.01, *n* = 10). Scale bars: 100 μm.

**Figure 11 ijms-23-07088-f011:**
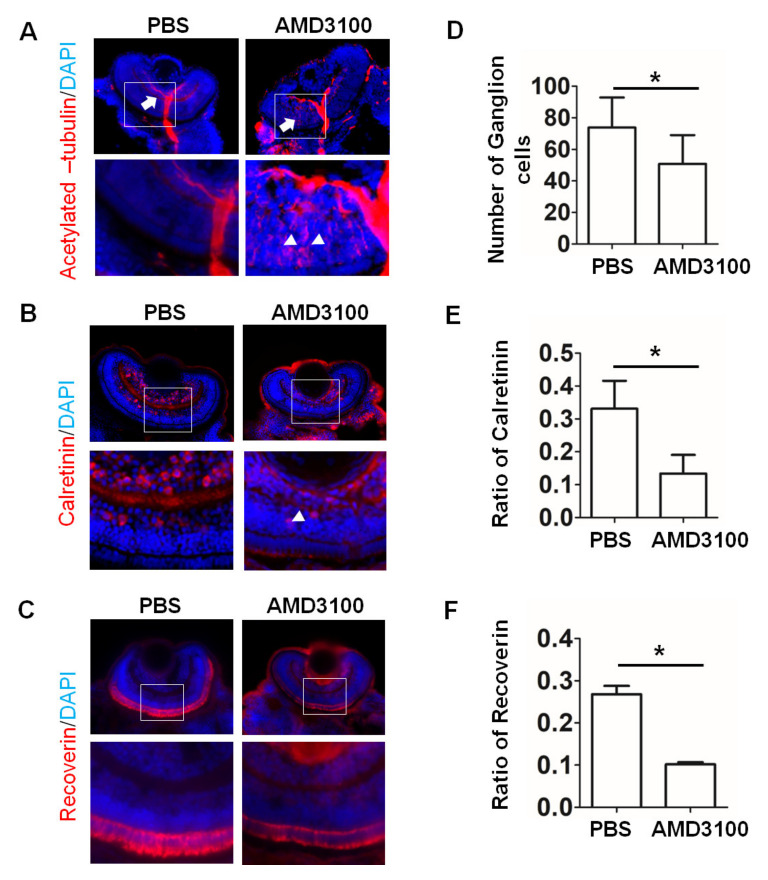
AMD3100 treatment in zebrafish leads to morphological changes in the retina similar to those detected in retinal organoids. (**A**) Ganglion cells were stained for acetylated tubulin in the PBS and AMD3100 treatment groups at 72 hpf, and ganglion cells were mislocalized in the AMD3100 treatment group (white arrow). (**B**) Amacrine cells were stained for calretinin in the PBS and AMD3100 treatment groups at 72 hpf, and amacrine cells were mislocalized in the AMD3100 treatment group (white arrow). (**C**) Photoreceptor cells were stained for Recoverin in the PBS and AMD3100 treatment groups at 72 hpf. (**D**) The relative ratio of acetylated tubulin-positive cells to all cells (nuclei were stained with DAPI) significantly decreased in the AMD3100 group (* *p* < 0.05, *n* = 12). (**E**) The relative ratio of calretinin-positive cells to all cells (nuclei were stained with DAPI) significantly decreased in the AMD3100 group (* *p* < 0.05, *n* = 12). (**F**) The relative ratio of the Recoverin-positive cells to all cells (nuclei were stained with DAPI) significantly decreased in the AMD3100 group (* *p* < 0.05, *n* = 12). Scale bars: 100 μm.

## Data Availability

The datasets generated during and/or analysed during the current study are not publicly available due to following study request, but are available from the corresponding author on reasonable request.

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
