# Peer review of "Comparison of the Response to the CXCR4 Antagonist AMD3100 during the Development of Retinal Organoids Derived from ES Cells and Zebrafish Retina"

_ijms, 2022, doi:10.3390/ijms23137088_

Round 1

Reviewer 1 Report

Wu et al described the effect of CXCR4 Antagonist AMD3100 in retinal organoids and zebrafish retinas. This is a short and descriptive paper, and several issues need to be addressed before this work can be published:

Major

The quality of the figures is very low. 

Fig 2. It is impossible to see the co-localization.

Line 130: Do the less transparent organoids have (more) RPE?  

Counting cell numbers with accessing proliferation and cell death makes it very difficult to draw conclusions. Authors should count ki67 positive cells and tunnel of cleaved caspase 3 cells. It would be interesting to see if the proliferating cells are already misplaced.

Figure 7:There is a clear rosette on the treated organoid, which suggests that the treatment is changing polarity. Authors should check for Muller glial cells and for OLM markers such as N-cadherin, PALS1, P120-catenin, etc, to see if the OLM integrity is also affected.

Figure 10. It is impossible to see the fibres.

Figure 11. As observed in the organoids also the zebrafish retinas have abnormal lamination, authors should explore and describe it in detail. 

Line 344. Lamination is heavily affected thus it seems difficult to say that these cells are not misplaced. 

Line 389: Authors do not show enough pieces of evidence to support this conclusion. As mentioned before is important to access proliferation, cell death and even possible cell cycle arrestment. 

Minor

Abstract: 

Line 16: It should be clear that the authors are referring to human organoids, that is not clear in the title nor in the abstract;

Line 18: "induced abnormal morphological changes." all changes are abnormal, the authors should rephrase the sentence;

Line 21: "consistent with those detected in organoids, except the photoreceptors did not show mislocalization", please mention the changes observed.

Line 22: "Therefore, our study suggests that retinal organoids may be a model for reproduction of retinal development; however, certain differences between organoids and the retina in vivo were detected." The sentence does not say anything useful. Please use the text to write the conclusion of the data obtained. 

Introduction

Line 41: "Many studies have demonstrated that retinal organoids are sufficient to simulate retinal development in vivo" What authors want to say with the sentence?

Line 54: Authors forgot to mention Muller glial cells.

Line 56: "But whether the retinal organoid was a reliable mimic model for exploring the genetic function in vitro was still not clear." Why is still not clear? Several papers showed great potential in mimicking disease models.

Line 60: please add a reference and specify "relative clear".

Line 115: It is hard to see the division between INL and ONL. 

Line 380-381: It is really the case? Both appear to have abnormal lamination, it is important to give an explanation why.  

Line 421: specify the NIM and RDM composition

Author Response

Wu et al described the effect of CXCR4 Antagonist AMD3100 in retinal organoids and zebrafish retinas. This is a short and descriptive paper, and several issues need to be addressed before this work can be published:

Major

  • The quality of the figures is very low. 

Fig 2. It is impossible to see the co-localization. 

Response:We are appreciated for the reviewer’s valuable suggestion, we have modified the Figure 2.

  • Line 130: Do the less transparent organoids have (more) RPE?  

Response:We appreciate the reviewer’s valuable suggestion. It has been reported RPE distributed in the peripheral area of the retinal organoid, while our results showed that the increasing non-transparent area in treated organoid was mainly located in the center (white arrow, Fig S3A, [1]). We further stained the organoid with MITF antibody (a marker for RPE cell), no MITF positive cell was detected inside the neural retina (Fig S3B). Therefore, the transparent alter was not associated with RPE. We presume the less transparent change might be associated with the cell apoptosis induced by CXCR4 blockage (Fig S4)

Reference:

[1] Zhong, X., Gutierrez, C., Xue, T., Hampton, C., Vergara, M. N., Cao, L. H., Peter, A., Park, T. S., Zambidis, E. T., Meyer, J. S. et al. Generation of three-dimensional retinal tissue with functional photoreceptors from human iPSCs. Nat. Commun, 2014, 5, 1-14.

  • Counting cell numbers with accessing proliferation and cell death makes it very difficult to draw conclusions. Authors should count ki67 positive cells and tunnel of cleaved caspase 3 cells. It would be interesting to see if the proliferating cells are already misplaced.

Response:We are appreciated for the reviewer’s valuable suggestion. According to the reviewer’s suggestion, we stained retinal organoids with antibodies of Ki67 and Cleaved-Caspase 3. The result demonstrated that the ratio of Ki67 positive cells were not different between the two groups. Meanwhile, the ratio of Ki67 positive cells mislocalization also did not show significantly different (white arrow). However, the ratio of Cleaved-Caspase 3 positive cells was significantly higher in 8w and 10w AMD3100 treatment retinal organoids. It implies that the CXCR4 blockage did not affect the cell proliferation, but promoted cell apoptosis cells in AMD3100 treatment organoid. The decrease of retinal neuron might cause by the cell apoptosis induced by CXCR4 blockage. We have revised and highlighted it in manuscript.

  • Figure 7: There is a clear rosette on the treated organoid, which suggests that the treatment is changing polarity. Authors should check for Muller glial cells and for OLM markers such as N-cadherin, PALS1, P120-catenin, etc, to see if the OLM integrity is also affected.

Response:We highly appreciated the reviewer’s valuable suggestion. The changing polarity might be induced by CXCR4 blockage due to CXCR4 bioactivity. We stained the 10w retinal organoids with CRALBP, a marker for Muller glial cells, but no CRALBP -positive cell detected in retinal organoid (data not show). The results is consisted with previous study that mature muller glial cells were not detected in early retinal organoid until 17w [1]. However, the organoids could not survive at that time after AMD3100 in present study. In addition, the OLM was composed by Muller cell in mature retinal organoid and involved in polar alter, but the specific cellular components of the OLM in early-staged retinal organoid remains unknown. Therefore, we will explore the association between the polarity-disturbed effect of CXCR4 signaling pathway and OLM integrity of immature retinal organoid in our future study.

Reference:

[1] Zhong, X., Gutierrez, C., Xue, T., Hampton, C., Vergara, M. N., Cao, L. H., Peter, A., Park, T. S., Zambidis, E. T., Meyer, J. S. et al. Generation of three-dimensional retinal tissue with functional photoreceptors from human iPSCs. Nat. Commun, 2014, 5, 1-14.

5)  Figure 10. It is impossible to see the fibres.

Response:We are sorry about our negligence, we have updated the Fig 10 in the manuscript.

  • Figure 11. As observed in the organoids also the zebrafish retinas have abnormal lamination, authors should explore and describe it in detail. 

Response: We are appreciated for the reviewer’s valuable suggestion. The AMD3100 treatment leaded to abnormal lamination in organoid and zebrafish retina, including the decrease of ganglion cells, amacrine cells, and photoreceptors and the distribution of neural out-growth, the misplace of ganglion cells, amacrine cells in zebrafish retina after AMD3100 treatment, which is consistent with previous study. CXCR4 played a key role in retinal cell migration, survive and nerve pathway in retina. CXCR4 deficiency would impair the normal retinal development, and the CXCR4 knockdown resulted in ganglion cell decrease and abnormal neural outgrowth in zebrafish retina [1, 2]. However, we did not detect the induction of abnormal photoreceptor ensembles by AMD3100 in the zebrafish retina.

We have modified and highlighted it in manuscript.

Reference:

[1] Li, Q., Shirabe, K., Thisse, C., Thisse, B., Okamoto, H., Masai, I., Kuwada, J. Y. et al. Chemokine signaling guides axons within the retina in zebrafish. J Neurosci, 2005, 25, 1711-1717.

[2] Chalasani, S. H., Baribaud, F., Coughlan, C. M., Sunshine, M. J., Lee, V. M., Doms, R. W., Littman, D. R., Raper, J. A. et al. The chemokine stromal cell-derived factor-1 promotes the survival of embryonic retinal ganglion cells. J Neurosci, 2003, 23, 4601-4612.

  • Line 344. Lamination is heavily affected thus it seems difficult to say that these cells are not misplaced. 

Response: We are appreciated for the reviewer’s valuable suggestion. Line 344, “Secondly, the blockage of CXCR4 lead to cellular mislocalization in retinal organoids and zebrafish, previous study demonstrated that the CXCR4 effected the cell migration and tissue polarity by interact with CXCL12 [40]”. So maybe the line number should be 279. Indeed, it was certainly hard to say the photoreceptors were not misplace at all. However, abnormal photoreceptor ensembles in zebrafish was not observed by Recoverin staining. We have modified and highlighted it in the discussion.

  • Line 389: Authors do not show enough pieces of evidence to support this conclusion. As mentioned before is important to access proliferation, cell death and even possible cell cycle arrestment. 

Response: We are appreciated for the reviewer’s valuable suggestion. We performed more experiments to support the conclusion by staining of Ki67 and cleaved Caspase 3 (Fig S3). We have modified and highlighted it in the discussion.

Minor

Abstract: 

Line 16: It should be clear that the authors are referring to human organoids, that is not clear in the title nor in the abstract

Response:We are appreciated for the reviewer’s valuable suggestion, we have revised and highlighted it in manuscript.

Line 18: "induced abnormal morphological changes." all changes are abnormal, the authors should rephrase the sentence;

Response:We are sorry about our negligence, we have rephrased and highlighted the sentence.

Line 21: "consistent with those detected in organoids, except the photoreceptors did not show mislocalization", please mention the changes observed.

Response:We are sorry about the negligence, we have revised and highlighted it in manuscript.

Line 22: "Therefore, our study suggests that retinal organoids may be a model for reproduction of retinal development; however, certain differences between organoids and the retina in vivo were detected." The sentence does not say anything useful. Please use the text to write the conclusion of the data obtained. 

Response:We are appreciated for the reviewer’s valuable suggestion, we have revised it in manuscript and highlighted.

Introduction

Line 41: "Many studies have demonstrated that retinal organoids are sufficient to simulate retinal development in vivo" What authors want to say with the sentence?

Response:We are sorry for our wrong description. It has modified to “the retinal organoids were considered to be a potential model to replace the in vivo model in retinal developmental research”.

Line 54: Authors forgot to mention Muller glial cells.

Response:We are appreciated for the review’s valuable suggestion. We have added and highlighted it in manuscript.

Line 56: "But whether the retinal organoid was a reliable mimic model for exploring the genetic function in vitro was still not clear." Why is still not clear? Several papers showed great potential in mimicking disease models.

Response:We are appreciated for the review’s valuable suggestion. There were papers showed great potential in retinal organoid mimicking disease model, but previous study mainly compared the morphological change and the key cell population that involve in the disease. There was little study compared the genetic function in various cell population in organoid and in vivo model, so the organoid was a reliable mimic model for exploring the genetic function in vitro was still not well defined. We have revised and highlighted it in the introduction.

Line 60: please add a reference and specify "relative clear".

Response:We are sorry about our negligence, the reference has been added and highlighted.

Line 115: It is hard to see the division between INL and ONL. 

Response:We are sorry about our negligence, the organoid only showed ganglion cell layer (GCL) and inner neuroblastic layer (INBL) at the developmental stage, the division between INL and ONL was still not clear. The photoreceptor cells located in the periphery of inner neuroblastic layer. We have modified it and highlighted in manuscript.

Line 380-381: It is really the case? Both appear to have abnormal lamination, it is important to give an explanation why.  

Response:We are appreciated for the review’s valuable suggestion. The lamination morphological changes seemed to be associated with the CXCR4 blockage. Previous studies have demonstrated that CXCR4 deficiency would impair the normal retinal development and CXCR4 knockdown resulted in abnormal lamination in zebrafish retina [1], which is consistent with our data. We have revised and highlighted it in the discussion.

Reference:

[1] Li, Q., Shirabe, K., Thisse, C., Thisse, B., Okamoto, H., Masai, I., Kuwada, J. Y. et al. Chemokine signaling guides axons within the retina in zebrafish. J Neurosci, 2005, 25, 1711-1717.

Line 421: specify the NIM and RDM composition

Response:We are sorry about the negligence, specifying has been added and highlighted in manuscript.

Reviewer 2 Report

In this paper the authors showed that inhibition of CXCR4 with the antagonist AMD3100 reduced the retinal organoid generation ratio and decreased the ratio of ganglion cells, amacrine cells and photoreceptors with mislocalization in retinal organoids. The comparison with zebrafish retina showed similar changes with retinal organoids after AMD3100 treatment.

Although the role of CXCR4 during development is well known, this study suggests that retinal organoids may be a suitable model to study retinal development. 

The overall writing and study design are good. The results generally merit to be published but I have some major and minor comments for the authors to clarify.

Major comments:

1) How did the authors confirmed the inhibition of CXCR4 after AMD3100 treatment? In the methods they mentioned that they assessed the concentration of AMD3100 by CCK-8 and migration assay but no indication of inhibition confirmation. 

2) I could find in many places in the paper discrepancy between the figure and the results obtained: in Figure 4 A, the pictures show a bigger MCM2 layer at 6w in the AMD treated organoids but the ratio MCM2/DAPI did not show a difference. In Figure 10C, the graph bars and their description in the results (Line 291) are not similar. In the discussion, line 345, the authors wrote that amacrine cells mislocalization was not observed in zebrafish but the results shown in Figure 11B indicate the opposite.

3) In Figure 3B, the authors observed that the organoids treated showed nontransparent NRs with increased in diameter. How the authors could explain these observations? even though they observed a decrease in ganglion, amacrine cells and photoreceptors after treatment.

4) Long term organoids present usually a necrotic core due to their thickness and insufficient nutrition infusion to the inside. This necrotic core was clearly visible in many 8w and 10w organoid pictures shown in this paper. This issue might have a great impact on the results, especially the Tuj1 and HuD-positive cells that are located in the center. How the authors addressed this?

5) The authors mainly used single staining to mark the retinal cell population. They used first HuD, Ap2a and OTX2 and then Brn3, calretinin and recoverin... Why the authors did not perform double staining with these markers to characterize their retinal cell population? Also, even though recoverin is a relatively mature photoreceptor marker the authors still indicate the positive cells as precursors (Figure 9).

6) In the zebrafish results, did the authors confirmed the CXCR4 expression?In addition, in Figure 11A the authors confirm the decrease of ganglion cells by counting the number of cells, in contrast to all data presented in the paper as ratio to DAPI. The authors might have a particular reason but was not addressed in the paper.  

Minor comments:

1) Line 95: The structure... quite similar... in vivo retina. It will be more clear to indicate specifically the similarities and discrepancy between both structures.

2) In Figure 1, the NEFL staining was not explained in the results paragraph 2.1.

3) In Figure 1A, the scheme of the retinal organoid protocol seems to end after 42 days. I suggest to include also phase contrast images of the organoids after 8w and 10w in culture.

Author Response

In this paper the authors showed that inhibition of CXCR4 with the antagonist AMD3100 reduced the retinal organoid generation ratio and decreased the ratio of ganglion cells, amacrine cells and photoreceptors with mislocalization in retinal organoids. The comparison with zebrafish retina showed similar changes with retinal organoids after AMD3100 treatment.

Although the role of CXCR4 during development is well known, this study suggests that retinal organoids may be a suitable model to study retinal development. 

The overall writing and study design are good. The results generally merit to be published but I have some major and minor comments for the authors to clarify.

Major comments:

  • How did the authors confirmed the inhibition of CXCR4 after AMD3100 treatment? In the methods they mentioned that they assessed the concentration of AMD3100 by CCK-8 and migration assay but no indication of inhibition confirmation. 

Response:We are appreciated for the reviewer’s suggestion. Every time, we confirmed the inhibition effect of the AMD3100 by cell migration assay because CXCR4 blockage inhibited the cell migration [1] because we had experience that AMD3100 did not work. Now the data was shown in supplemental data S1.

Reference:

[1] Liu J M, Zhao K, Du L X, et al. AMD3100 inhibits the migration and differentiation of neural stem cells after spinal cord injury. Sci Rep, 2017, 7, 1-9.

2) I could find in many places in the paper discrepancy between the figure and the results obtained: in Figure 4 A, the pictures show a bigger MCM2 layer at 6w in the AMD treated organoids but the ratio MCM2/DAPI did not show a difference. In Figure 10C, the graph bars and their description in the results (Line 291) are not similar. In the discussion, line 345, the authors wrote that amacrine cells mislocalization was not observed in zebrafish but the results shown in Figure 11B indicate the opposite.

Response: The ratio MCM2/DAPI was calculated according to more than 10 orgnoids. Indeed, we check the ratio of MCM2/DAPI and make sure it did not show a difference between the PBS and AMD3100 treatment group. We are sorry about our negligence during organizing the manuscript for using an inappropriate image to show our result in Fig. 4A, we have replaced the image in Fig 4.

In Fig.10C, sorry for our mistake. The figure showed the statistical graph indicating the ratio of NEFL mislocalization to the NEFL staining, while the results described the ratio of NEFL mislocalization to the DAPI staining. To consistent with the other data presented in our study, we preferred to present the ratio of mislocalilzation to the DAPI staining in Fig. 10C. Meanwhile, the SD value in 8w AMD3100 treatment should be 0.044. We have replaced the graph in Fig 10C.

In the discussion, the amacrine cells mislocalization was certainly observe in zebrafish, the sentence has been revised and highlighted.

3) In Figure 3B, the authors observed that the organoids treated showed nontransparent NRs with increased in diameter. How the authors could explain these observations? Even though they observed a decrease in ganglion, amacrine cells and photoreceptors after treatment.

Response:We are appreciated for the review’s question, we further compared the thickness of NRs in PBS and AMD3100 treatment groups, the result showed that the thickness of NRs was thinner in AMD3100 group. Nontransparent area could be associated with apoptotic cells, which was consistent with the decrease in ganglion, amacrine cells and photoreceptors. The increasing in diameter was caused by cells abnormal arrangement due to CXCR4 blockage.

4) Long term organoids present usually a necrotic core due to their thickness and insufficient nutrition infusion to the inside. This necrotic core was clearly visible in many 8w and 10w organoid pictures shown in this paper. This issue might have a great impact on the results, especially the Tuj1 and HuD-positive cells that are located in the center. How the authors addressed this?

Response:We are appreciated for the review’s question, long term organoid certainly leaded to necrotic core due to the thickness and insufficient nutrition infusion to the inside. However, fewer Cleaved-Caspase 3-positive cells was observed in organoids with or without AMD3100 treatment (0.1-0.2%), whereas, the ratio of TUJ1 and HuD-positive cells is more than 20% (Fig 4 and Fig 5). Thus, the necrotic core might not cause the change in our study.

5) The authors mainly used single staining to mark the retinal cell population. They used first HuD, Ap2a and OTX2 and then Brn3, calretinin and recoverin... Why the authors did not perform double staining with these markers to characterize their retinal cell population? Also, even though recoverin is a relatively mature photoreceptor marker the authors still indicate the positive cells as precursors (Figure 9).

Response: We are appreciated for the reviewer’s question. Double staining was certainly a better choice to characterize the retinal population more specific. Sorry for our defeat in experimental design. Most of our antibodies were rabbit IgG, it was difficult to used double staining to characterize the retinal cell population. To cover the shortage, we double stained the ganglion cell with HuD and TUJ1 antibodies in 8w organoids, the result was showed below. HuD (nuclei) and TUJ1 (cytoplasm) presented colocation. Similar to the single staining, CXCR4 blockage resulted in decrease and mislocalization of ganglion cells which was consistent with the single staining. Thus, the single staining might not affect the final conclusion.

Recoverin was certainly a relatively mature photoreceptor marker in previous paper. Because the more mature photoreceptor marker rhodopsin and S-opsin was not expressed in the 10w organoid [2], we chose Recoverin as the photoreceptor marker in our study. It was consistent with the previous study, the rhodopsin and S-opsin positive present by 21w to 22w [2].

Reference:

[1] Tang M, Luo Z, Wu Y, et al. BAM15 attenuates transportation-induced apoptosis in iPS-differentiated retinal tissue. Stem Cell Res Ther, 2019, 10, 1-11.

[2] Zhong, X., Gutierrez, C., Xue, T., Hampton, C., Vergara, M. N., Cao, L. H., Peter, A., Park, T. S., Zambidis, E. T., Meyer, J. S. et al. Generation of three-dimensional retinal tissue with functional photoreceptors from human iPSCs. Nat. Commun, 2014, 5, 1-14.

6) In the zebrafish results, did the authors confirmed the CXCR4 expression? In addition, in Figure 11A the authors confirm the decrease of ganglion cells by counting the number of cells, in contrast to all data presented in the paper as ratio to DAPI. The authors might have a particular reason but was not addressed in the paper.  

Response: Thank you for your valuable suggestion. The CXCR4 expression in zebrafish retina was showed in Fig.S5. The result showed that CXCR4 most located in the ganglion cell layer of zebrafish which was consistent with the previous paper [1].

We confirmed the decrease of ganglion cells by counting the cells instead of ratio of DAPI because Acetylated –tubulin expressed in cytoplasm and stained the optic nerve in zebrafish retina. The ratio of Acetylated –tubulin would be less accurate than the cell counting to present the change in ganglion cell number.

Reference:

[1] Li, Q., Shirabe, K., Thisse, C., Thisse, B., Okamoto, H., Masai, I., Kuwada, J. Y. et al. Chemokine signaling guides axons within the retina in zebrafish. J Neurosci, 2005, 25, 1711-1717.

Minor comments:

  • Line 95: The structure... quite similar... in vivo retina. It will be more clear to indicate specifically the similarities and discrepancy between both structures.

Response: We are appreciated for the reviewer’s suggestion, and we have indicated the similarities and discrepancy between both structures in manuscript and highlight.

  • In Figure 1, the NEFL staining was not explained in the results paragraph 2.1.

Response:We are sorry for our negligence, the explain about the NEFL has been added and highlighted.

3) In Figure 1A, the scheme of the retinal organoid protocol seems to end after 42 days. I suggest to include also phase contrast images of the organoids after 8w and 10w in culture.

Response: We are appreciated for the reviewer’s suggestion, and we have added the images of the 8w and 10w organoids in Figure 1A.

Round 2

Reviewer 1 Report

The authors have replied to all my comments. The manuscript is now ready fo publication

Reviewer 2 Report

The authors addressed my major and minor comments. I support now the revised manuscript for publication.